# Cellular SPION Uptake and Toxicity in Various Head and Neck Cancer Cell Lines

**DOI:** 10.3390/nano11030726

**Published:** 2021-03-13

**Authors:** Matthias Balk, Theresa Haus, Julia Band, Harald Unterweger, Eveline Schreiber, Ralf P. Friedrich, Christoph Alexiou, Antoniu-Oreste Gostian

**Affiliations:** 1Department of Otorhinolaryngology, Head and Neck Surgery, Section of Experimental Oncology and Nanomedicine (SEON), Else Kröner-Fresenius-Stiftung Professorship, Universitätsklinikum Erlangen, 91054 Erlangen, Germany; Matthias.Balk@uk-erlangen.de (M.B.); Theresa.Haus@gmx.de (T.H.); Julia.Band@uk-erlangen.de (J.B.); Harald.Unterweger@uk-erlangen.de (H.U.); Eveline.Schreiber@uk-erlangen.de (E.S.); Christoph.Alexiou@uk-erlangen.de (C.A.); Antoniu-Oreste.Gostian@uk-erlangen.de (A.-O.G.); 2Friedrich-Alexander-Universität Erlangen-Nürnberg (FAU), 91054 Erlangen, Germany

**Keywords:** superparamagnetic iron oxide nanoparticles, cytotoxicity, flow cytometry, atomic emission spectroscopy, holotomographic microscopy

## Abstract

Superparamagnetic iron oxide nanoparticles (SPIONs) feature distinct magnetic properties that make them useful and effective tools for various diagnostic, therapeutic and theranostic applications. In particular, their use in magnetic drug targeting (MDT) promises to be an effective approach for the treatment of various diseases such as cancer. At the cellular level, SPION uptake, along with SPION-mediated toxicity, represents the most important prerequisite for successful application. Thus, the present study determines SPION uptake, toxicity and biocompatibility in human head and neck tumor cell lines of the tongue, pharynx and salivary gland. Using magnetic susceptibility measurements, microscopy, atomic emission spectroscopy, flow cytometry, and plasma coagulation, we analyzed the magnetic properties, cellular uptake and biocompatibility of two different SPION types in the presence and absence of external magnetic fields. Incubation of cells with lauric acid and human serum albumin-coated nanoparticles (SPION^LA-HSA^) resulted in substantial particle uptake with low cytotoxicity. In contrast, uptake of lauric acid-coated nanoparticles (SPION^LA^) was substantially increased but accompanied by higher toxicity. The presence of an external magnetic field significantly increased cellular uptake of both particles, although cytotoxicity was not significantly increased in any of the cell lines. SPIONs coated with lauric acid and/or human serum albumin show different patterns of uptake and toxicity in response to an external magnetic field. Consequently, the results indicate the potential use of SPIONs as vehicles for MDT in head and neck cancer.

## 1. Introduction

Squamous cell carcinoma of the head and neck is the sixth most common malignant tumor in the world, with 500,000 new cases every year [1]. In particular, alcohol and tobacco consumption, as well as the human papilloma virus, are among the main causes of these head and neck tumors [2,3]. Therapy usually includes ablative surgical procedures and/or chemoradiotherapy, especially for advanced tumor stages [4]. Notwithstanding the emerging era of immunotherapy, cisplatin remains the standard cytostatic agent in the first-line and second-line treatment of advanced head and neck tumors [5,6,7]. However, in addition to its efficacy, cisplatin also causes potentially severe and thus therapy-limiting systemic side effects, e.g., about one-third of patients develop acute renal failure after high-dose cisplatin administration [3,8,9]. A potentially effective way to circumvent the systemic toxicity of cytostatic drugs is to develop targeted drug delivery methods [10]. In particular, nanomaterials have a high potential to serve as diagnostic tools and efficient drug delivery systems to target various cancers [11,12,13,14,15,16]. In this regard, magnetic drug targeting (MDT) has already demonstrated its efficacy and feasibility to accumulate chemotherapeutic agents in the targeted tumor lesions, thereby enhancing the therapeutic effect while reducing adverse drug effects [17,18,19]. In past studies, superparamagnetic iron oxide nanoparticles (SPIONs) functionalized with chemotherapeutic agents were successfully directed into tumor regions via external magnetic fields in animal models, confirming the in vivo applicability of MDT [20]. In the aforementioned study, more than half of the cytostatic drug mitoxantrone (MTO) could be accumulated in the tumor region after intra-arterial application of MTO-functionalized SPION compared with less than 1% after conventional systemic intravenous administration. In addition, MDT also reduced the renal load of the cytostatic drug by more than 60%. However, the application of MDT using SPIONs in head and neck tumors has not received much attention to date. The work presented here focuses on the biological impact of SPIONs coated with lauric acid (SPION^LA^) and SPIONs coated with lauric acid and human serum albumin (SPION^LA-HSA^) on various head and neck cancer cell lines. Accordingly, the primary objective of the present work was to investigate the effects of both SPION systems on viability, cell proliferation and uptake in head and neck tumor cell lines of squamous cell carcinoma of the pharynx, tongue and parotid gland. In addition, the influence of magnetically enhanced SPION accumulation, as a prerequisite for their suitability as potential vehicles for MDT, was investigated by culturing the cells either without or on magnetic plates. Furthermore, biocompatibility with human blood representing the requirement for future in vivo applications served as secondary objective and was analyzed using coagulation assays with human blood plasma.

## 2. Materials and Methods

### 2.1. Materials

Iron (II) chloride tetrahydrate (FeCl_2_⋅4H_2_O), iron (III) chloride hexahydrate (FeCl_3_⋅6H_2_O) and Sartorius ultrafiltration tubes Vivaspin 20, PES with a molecular weight cut-off (MWCO) of 100 kDa were purchased from VWR, Darmstadt, Germany. Iron reference standards were bought from Bernd Kraft GmbH, Duisburg, Germany. Glass coverslips, sterile Rotilabo^®^ syringe filters with cellulose mixed ester membranes, dialysis bags (Repligen, Spectra/Por 6, MWCO 10 kDa), nitric acid, DMSO, trypsin, PBS, ethanol, disodium hydrogen phosphate, gelatin and formalin were supplied by Roth, Karlsruhe, Germany. Propidium iodide (PI), Triton X-100, lauric acid and Ribonuclease A were purchased from Sigma-Aldrich, St. Louis, MO, USA. Anexin V-FITC (AxV-FITC), Hoechst 33342, Alexa Fluor 488 Phalloidin and 1,1′,3,3,3′,3′-hexamethylindodicarbocyanine iodide (DiIC1(5)) were purchased from Life Technologies, Darmstadt, Germany. Ringer’s solution were purchased from Fresenius Kabi AG, Bad Homburg, Germany. Fetal calf serum (FCS) superior, were purchased from Thermo Fisher Scientific, Waltham, MA, USA. EMEM, DMEM: F-12 and McCoy’s 5A Medium were purchased from ATCC^®^, Manassas, VA, USA. Hydrocortisone was purchased from Pfizer, New York City, NY, USA. Ferucarbotran (Resovist^®^) were purchased from I’rom Group, Tokyo, Japan and Ferumoxytol (Rienso^®^) was purchased from Takeda Pharmaceutical Company Limited, Osaka, Japan. Water used in all experiments was of ultrapure quality produced by the Milli-Q^®^ system from Merck, Darmstadt, Germany. Other materials and equipment used in this study includes: Human serum albumin (HSA) solution (Recombumin Elite, 10% *w/v*, Albumedix Nottingham, UK); Agilent 4200 microwave plasma-atomic emission spectrometer (MP-AES) (Agilent Technologies, Santa Clara, CA, USA); Malvern Zetasizer (Malvern Instruments, Worcestershire, England); MS2G Susceptometer, Bartington^®^ Instruments, Oxon, UK); aPTT Kit and 25 mM CaCl_2_ solution (DiaSys Deutschland Vertriebs-GmbH, Flacht, Germany); MC4 plus plasma coagulometer (Merlin medical, Lemgo, Germany); Magnetic plates (Fa. Karl Lettenbauer, Erlangen, Germany); S-15-08-N magnets (Webcraft GmbH, Gottmadingen, Germany); FaDu (ATCC^®^ HTB-43™), SCC-9 (ATCC^®^ CRL-1629™), A-253 (ATCC^®^ HTB-41™) and Detroit 562 (ATCC^®^ CCL-138™) cell lines (ATCC^®^, Manassas, VA, USA); Gallios cytofluorometer and Kaluza software version 2.0 (Beckman Coulter, Fullerton, CA, USA); Incucyte^®^ (Sartorius, Göttingen, Germany); xCELLigence (Agilent, Santa Clara, CA, USA); Mounting Medium (Dako North America, Carpinteria, CA, USA); Zeiss Axio Observer. Z1 fluorescent microscope (Carl Zeiss AG, Oberkochen, Germany); Nanolive Fluo-3D Cell Explorer^®^ and Steve software v.1.6.3496^®^ (NanoLive, Tolochenaz, Switzerland), Thermal mixer (Eppendorf, Hamburg, Germany), KrosFlo^®^ Tangential flow ultrafiltration equipment (Repligen, Rancho Dominguez, CA, USA).

### 2.2. Synthesis of Lauric Acid-Coated Iron Oxide Nanoparticles (SPION^LA^)

SPION^LA^ was synthesized according to the protocol of Zaloga et al., 2014 [21]. Briefly, Fe (II) chloride and Fe (III) chloride were dissolved in ultrapure water and co-precipitated under stirring at 80 °C in alkaline media under an argon atmosphere. Subsequently, the particles were coated with lauric acid. The suspension was homogenized for 30 min at 90 °C, then dialyzed several times (MWCO of 10 kDa) and dissolved in ultrapure water.

### 2.3. Synthesis of Lauric Acid- and Human Serum Albumin-Coated Iron Oxide Nanoparticles (SPION^LA-HSA^)

The coating of SPION^LA^ particles with human serum albumin (HSA) was performed according to a protocol of Zaloga et al., 2016 [22]. Briefly, using a sterile filter (0.22 µm), human serum albumin (HSA) solution was transferred into dialysis bags (MWCO 10 kDa) and dialyzed against 4.5 L of ultrapure water (4 water changes, 5 h). Subsequently, the solution was concentrated by tangential flow ultrafiltration as previously described [23]. The HSA solution was stirred at room temperature and SPION^LA^ was added dropwise through a filter (0.8 µm). Excess HSA was removed by tangential flow ultrafiltration and the suspension was filtered through a sterile filter (0.22 µm).

### 2.4. Iron Quantification of Nanoparticles

The iron content of the of both particle types was determined using an Agilent 4200 microwave plasma-atomic emission spectrometer (MP-AES). 50 µL of the aqueous diluted suspension was dissolved in 50 µL 65% HNO_3_ and incubated for 10 min at 95 °C. The sample was then further diluted in 1900 µL ultrapure water and measured with MP-AES. A commercial iron solution was used as external standard.

### 2.5. Dynamic Light Scattering (DLS) and Zeta Potential Measurements

A Malvern Zetasizer was used to determine the hydrodynamic size of the particles in water, FCS-free medium (EMEM) and medium with 10% FCS. The particles were diluted to an absolute iron content of 50 µg/mL and measured in triplicate at 25 °C.

### 2.6. Magnetic Susceptibility Measurements 

As measure of the magnetizability, the magnetic susceptibility of the particles was determined using a commercial MS2G susceptometer and normalized to the iron content of the suspension. Similarly, the magnetic susceptibility of Resovist^®^ and Rienso^®^ was analyzed as a reference.

### 2.7. Plasmacoagulation Assay

The coagulation of plasma in presence of particles was investigated using 10 mL freshly drawn human citrate blood samples and the aPTT Kit with BasiCon1 and BasiCon2 as controls. The blood samples were centrifuged 15 min at 2500 rcf and the plasma-containing supernatant was separated from the sedimented cells. 30 µL of particle dilutions in the concentration 500, 1000 and 1500 µg/mL were added to 270 µL citrate stabilized plasma and incubated for 30 min at 37 °C. Using the MC4 plus plasma coagulometer, 100 µL plasma-particle suspension were mixed with 100 µL aPTT solution and incubated for 2 min. The coagulation time was automatically monitored after adding 100 µL 25 mM CaCl_2_. The use of human blood was approved by the local ethics committee at Universitätsklinikum Erlangen (reference no. 257-14 B).

### 2.8. Design of the Magnetic Plates

Magnetic plates are exerting a maximum magnetic force of 380 mT on the bottom of the cell culture plates. The 3-layer construction of the magnetic plates in the dimension 124 mm × 82.8 mm have a total thickness of 14 mm and consists of a top plate made of 8.1 mm flat milled “Plexiglas GS” with 4 × 6 holes with a diameter of 15.5 mm for round magnets (15 mm × 8 mm, S-15-08-N, Neodym, magnetization N42, holding force approx. 6.2 kg (=60.8 N), a middle plate made of 2 mm magnetic v2a sheet metal, drilled and countersunk for screwing to the upper plate and a bottom plate of 4 mm polyoxymethylene (POM) with deeply countersunk mounting holes for screwing to the middle plate and the top plate (Appendix A).

### 2.9. Cell Culture Conditions

Four human carcinoma cell lines from the head and neck area were purchased from ATCC^®^. FaDu is a cell line derived from a primary hypopharyngeal tumor, SCC-9 is derived from a primary tumor of the tongue, A-253 was isolated from a primary submaxillary salivary gland tumor and Detroit 562 was isolated from a metastasis of a pharyngeal tumor. Cells were cultivated according to manufacturer’s instructions.

### 2.10. Cellular SPION Toxicity and SPION Uptake by Flow Cytometry

To measure the cellular toxicity, uptake and attachment of the different SPIONs, cells were seeded in 6-well plates in a total volume of 2.7 mL per well. After a confluence of ≈60% was reached, 300 µL SPION^LA^ or SPION^LA-HSA^ were added at total iron concentrations of 50 µg/mL, 100 µg/mL and 150 µg/mL, respectively. Instead of SPION solutions, the negative control contained the same amount of H_2_O and the toxicity control DMSO (final DMSO concentration 2%). After SPION addition, samples were further incubated either in the absence of an external magnetic field or on magnetic plates. Controls were incubated in absence of magnetic fields. After an incubation period of 48 h, the cells were harvested using trypsin. Subsequently, the samples containing the harvested cells, all media and washing solutions, were centrifuged at 300 rcf for 5 min and washed twice with 3 mL phosphate-buffered saline (PBS). After the last centrifugation step, the pellet was resuspended in 0.5 mL PBS. The cell suspension were used for the subsequent analysis by flow cytometry and MP-AES. The effects of SPIONs on cellular cytotoxicity and granularity was determined by flow cytometry using a Gallios cytofluorometer [24,25]. To measure cell viability and granularity, 40 µL cell suspension were incubated with 250 µL freshly prepared staining solution containing 10 pg/mL AxV-FITC, 10 µg/mL Hoechst 33342, and 2.04 µg/mL DiIC1(5), 66.6 ng/mL propidium iodide (PI) in Ringer’s solution for 20 min at 4 °C [24]. To determine the DNA status, 200 µL of the cell suspension were fixed with 3 mL 70% ice-cold ethanol and stored at −20 °C for at least one hour. After centrifugation (5 min at 400 rcf), the cells were washed once with PBS and then absorbed into 0.5 mL PBS. 0.5 mL DNA extraction buffer (192 mL 0.2 M Na_2_HPO_4_, 8 mL Triton X-100 (0.1%) was added and incubated for 5 min at room temperature. After centrifugation (5 min at 400 rcf) the supernatant was aspirated to 0.1 mL and incubated with 0.4 mL staining solution (20 µg/mL PI and 20 µg/mL Ribonuclease A in PBS) for 30 min dark at room temperature. The amount of cellular nanoparticles was determined by changes in granularity represented by alteration in the side scatter (SSc) intensity. Increases in SSc caused by apoptotic and necrotic processes were excluded by gating on viable cells. All flow cytometry analyses were conducted in four independent experiments with triplicates. Electronic compensation was used to eliminate bleed through fluorescence. Data analysis was performed with Kaluza software version 2.0.

### 2.11. Cellular SPION Amount Measured by MP-AES

200 µL of the cell suspension was centrifuged for 5 min at 5000 rcf. After drying the cell pellets at 95 °C for 30 min in a thermal mixer, the pellets were dissolved with 50 µL HNO_3_ (65%) for 15 min at 95 °C and then diluted with 950 µL ultrapure water. The iron content of the sample was determined by MP-AES and normalized to the cell numbers.

### 2.12. Cell Proliferation Monitored by Live Cell Imaging and Impedance-Based Technology

Growth curves were determined by Incucyte^®^ and xCELLigence. In short, the cells were seeded in 96-well plates in a volume of 180 µL. After 24 h incubation time, 20 µL SPION solution were added to achieve total SPION concentrations of 50 µg/mL, 100 µg/mL and 150 µg/mL. Additionally, the same amount of H_2_O and DMSO (final concentration 2%) served as controls. The growth of the cells were recorded over seven days by live cell imaging with IncuCyte^®^ and with the impedance based technology from xCELLigence^®^. The experiments were performed in three independent experiments with eight replicates each.

### 2.13. Cellular SPION Uptake Visualized by Fluorescence Microscopy

For fluorescence imaging of SPION-treated cells, 24-well plates were equipped with gelatin-coated glass coverslips before cells were seeded in a volume of 900 µL. After a confluence of ≈60% was reached, 100 µL SPIONs were added at a final concentration of 100 µg/mL. H_2_O and DMSO (final concentration 2%) served as controls. Samples were incubated for 48 h with and without magnetic plates. Cells were washed with PBS and fixed with 0.5 mL formalin (4% in PBS) for 30 min. After another washing step with PBS, the cells were permeabilized with 0.5 mL TritonX-100 (0.2% in PBS) for 3 min. After washing twice with PBS, Alexa Fluor 488 Phalloidin (1:100) and Hoechst 3334 (1:2000) were added to the samples and cells were incubated for another 30 min. Subsequently, the cells were washed twice with PBS and embedded in Mounting Medium. Cells were imaged with a Zeiss Axio Observer. Z1 fluorescent microscope.

### 2.14. Holotomographic Imaging

For three dimensional (3D) imaging by holotomography, 5000 cells were seeded into 32 mm µ-Dishes (Ibidi, Gräfelfing, Germany) in a volume of 2.7 mL. After 24 h, 300 µL SPION^LA^ and SPION^LA-HSA^ were added to a final iron concentration of 100 µg/mL, whereas control samples were supplemented with the same amount of ultrapure water. Cells were incubated for another 48 h, with half of the samples placed on a magnetic plates. After washing with PBS, cells were imaged using the live cell 3D holotomographic microscop Nanolive Fluo-3D Cell Explorer^®^ [26]. Steve software v.1.6.3496^®^ was used for digital staining, 3D rendering and data export. Digital staining of high refractive index (RI) in red representing nanoparticle clusters in particular, and low RI in green representing cellular structures such as cytoplasm, nuclei etc., allowed quantification of the cellular volume of high RI and thus the determination of nanoparticle clusters. From each sample, the RI of at least 30 images with a total of 100–250 single cells were analyzed. The RI volumes of each image were then normalized to the cell number and data sets were analyzed using Microsoft Excel.

## 3. Results

### 3.1. Physicochemical SPION Characterization

Since the cellular uptake and biocompatibility of SPIONs are primarily determined by their intrinsic properties, we first analyzed the hydrodynamic size and zeta potential using dynamic light scattering (DLS) (Table 1, Appendix A). The zeta potential of SPION^LA^ and SPION^LA-HSA^ in ultrapure water showed a negative value of −35.0 mV and −20.3 mV, respectively. The suitability of the particles for further in vitro experiments was ensured by measuring the hydrodynamic diameters not only in ultrapure water but also in cell culture medium containing 0% FCS and in medium containing 10% FCS. In water, the particles exhibited hydrodynamic sizes of 46 nm (SPION^LA^) and 69 nm (SPION^LA-HSA^) without the appearance of larger aggregates, indicating very high stability. The slight size increase found for SPION^LA-HSA^ particles indicates the successful formation of a HSA protein corona. In addition, when diluting SPION^LA-HSA^ in cell culture media, regardless of the presence or absence of FCS, the size was not strongly altered. The measured decrease in numerical value from 69 nm to 50 nm in media containing 10% FCS is most likely due to a forced leftward shift of the size curves caused by small protein particles (11 nm) within the FCS (Appendix A). In contrast to SPION^LA-HSA^, SPION^LA^ agglomerated in media without FCS, resulting in a size of about 2.5 µm. However, the colloidal stability is restored in FCS-containing media, indicating the formation of a stabilizing protein corona and confirming the suitability of both particle systems for subsequent in vitro experiments. Magnetically assisted SPION uptake is dependent on the magnetizability of the particles. Compared to the magnetic susceptibility of Resovist^®^ (7.72 × 10^−3^) and the relatively low value of Rienso^®^ (1.49 × 10^−3^), SPION^LA^ (7.27 × 10^−3^) and SPION^LA-HSA^ (7.37 × 10^−3^) demonstrated magnetic properties very similar to the well-known contrast agent Resovist^®^ which has been shown to be suitable not only for magnetic resonance imaging but also for regenerative medicine and MDT [27,28,29]. 

### 3.2. Cellular SPION Uptake Visualized by Fluorescent Microscopy

Differences in magnetically enhanced particle uptake were first analyzed using bright-field and fluorescence microscopy. We explicitly investigated cellular uptake efficiency without fluorescent SPION-labelling, as this affects the physicochemical properties of the particles and their binding behavior to cells [30]. Although light microscopy only allows acquisition of 2D images, the increase in high contrast vesicles and their perinuclear localization pattern in images taken by differential interference contrast (DIC) and phase contrast suggests an increase in cellular or cell-associated particles. (Figure 1, Appendix A). Incubation with SPION^LA^ or SPION^LA-HSA^ resulted in a clear increase in cellular nanoparticle uptake or binding. In general, SCC-9 and A-253 cells (Figure 1, Appendix A) showed a higher amount of cellular particles than Detroit 562 and FaDu cells (Appendix A), indicating cell line-dependent differences in the efficiency of SPION uptake. In addition, applying a magnetic field enhanced the SPION uptake more clearly in SCC-9 and A-253 cells than in the other two cell lines. Moreover, compared with SPION uptake in Detroit 562 and FaDu cells, differences in cellular uptake between SPION^LA^ and SPION^LA-HSA^ were more pronounced in SCC-9 and A-253 cells, suggesting particle- and cell-dependent uptake efficiency.

### 3.3. Cellular SPION Amount Determined by Holotomography 

Recently, we demonstrated that cellular uptake of fluorescently unlabeled nanoparticles can be reliably monitored and quantified by holotomographic imaging [26]. Here, this technique was utilized to further confirm the data obtained by optical microscopy. After addition of SPION^LA^ and SPION^LA-HSA^ to the cell culture, Petri dishes were incubated for 48 h, with half of the samples placed on a magnetic plates. Cells were imaged by 3D refractive index scanning and digital staining of high RI in reddish-brown representing nanoparticle clusters in particular, and low RI in green representing cellular structures such as cytoplasm, nuclei etc., and allowed quantification of nanoparticle clusters. Figure 2 and Appendix A clearly shows enrichment of both SPION^LA^ and SPION^LA-HSA^ in the presence of a magnetic field. In addition, the images indicate a substantially stronger accumulation of SPION^LA^ compared to SPION^LA-HSA^. These observations were confirmed after quantification of the RI signal, which also shows that the amount of cellular SPION^LA^ is significantly higher than that of SPION^LA-HSA^ (Figure 3).

### 3.4. Cellular SPION Quantification by Flow Cytometry

The presence and accumulation of cellular particles were further validated by flow cytometric side scatter (SSc) analysis in viable cells (Figure 4) [25]. Furthermore, SSc analysis provided additional data on particle- and cell line-dependent differences in the efficiency of cellular particle accumulation. All cell lines showed a significant dose-dependent increase in SSc intensity. In addition, the SSc increase was significantly higher at concentrations above 100 µg_Fe_/mL when cells were incubated with SPION^LA^ compared with SPION^LA-HSA^, reaching up to a 2.5-fold SSc increase at a SPION concentration of 150 µg_Fe_/mL. Moreover, the SSc increase was higher when SPION-treated cells were incubated on a magnet, indicating magnetically forced cellular accumulation. Thus, the findings are in very good agreement with the trends identified in the analysis of holotomographic microscopy images (Figure 3). This becomes even more evident when comparing the quantification of holotomographic images of cells incubated with 100 µg_Fe_/mL with SSc results obtained with the same SPION concentration. Thus, analyses of morphological changes in SPION-treated cells using light scattering and refractive index-based technologies provide concordant results for the presence of cellular SPIONs.

### 3.5. Cellular SPION Quantification by Atomic Emission Spectroscopy

A reliable and precise quantification of elemental iron and thus determination of the amount SPIONs can be performed using highly sensitive spectroscopic methods, such as atomic absorption spectroscopy (AAS) and atomic emission spectroscopy (AES) [25]. We therefore incubated the head and neck cancer cell lines with different concentrations of SPION^LA-HSA^ or SPION^LA^, either in presence (+) or in absence (−) of a magnetic force. We then analyzed the cellular iron content by microwave plasma-atomic emission spectrometry (MP-AES) as well as compared the data with the results of holotomography and flow cytometry (Figure 5). Without magnetic force, the maximum cellular iron amount was below 2 pg_Fe_/cell in all cell lines after incubation with SPION^LA-HSA^. After incubation of cells on magnetic plates, the cellular iron amount was significantly higher, reaching up to 11.7, 7.0, 28.8, and 15.3 pg_Fe_/cell in FaDu, Detroit-562, A-253 and SCC-9 cells, respectively. Compared to SPION^LA-HSA^, cellular uptake of SPION^LA^ was clearly increased even without incubation on magnetic plates. Without magnets, incubation of FaDu, Detroit-562, A-253 and SCC-9 cells with 100 µg_Fe_/mL SPION^LA^ resulted in a cellular iron loading of 3.0, 2.2, 5.1, and 8.1 pg_Fe_/cell, respectively, and with the magnetically enhanced approach 17.7, 14.3, 42.3, and 104.3 pg_Fe_/cell, respectively. Thus, using MP-AES, we confirmed a dosage-dependent cellular SPION uptake or binding and a significantly higher uptake efficiency of SPION^LA^ compared with SPION^LA-HSA^. Furthermore, cellular SPION content was significantly increased in cells incubated in the presence of an external magnet compared to samples incubated without a magnet.

### 3.6. Cellular SPION Toxicity

After the extensive investigation of cellular SPION uptake, we used flow cytometry to determine the influence of SPION^LA^ and SPION^LA-HSA^ on viability, necrosis, apoptosis, mitochondrial membrane integrity, DNA fragmentation, and cell cycle (Figure 6, Appendix A) [31,32,33]. Determination of the amount of viable, necrotic, and apoptotic cells after treatment with both SPION types showed a clear dose-dependent decrease in viability for all cell lines (Figure 6). Of note, SPION^LA^ had a comparably greater detrimental effect on cell viability compared to SPION^LA-HSA^. Interestingly, despite the strong increase in SPION uptake in the presence of a magnetic field (Figure 3, Figure 4 and Figure 5), only FaDu cells showed a slight decrease in viability, whereas the viability of Detroit 562, A-253, and SCC-9 cells did not deteriorate in the presence of external magnets. 

The lower cytotoxicity of SPION^LA-HSA^ compared with SPION^LA^ was confirmed by mitochondrial membrane potential (mmp) measurements using flow cytometry after DiIC1(5) and Hoechst staining (Appendix A). All cell lines showed a dose-dependent increase in the number of cells with impaired mmp. Again, the changes in mmp were more pronounced after incubation with SPION^LA^ than after SPION^LA-HSA^ treatment, and the influence of external magnets was detectable only in FaDu cells and not in Detroit 562, A-253 and SCC-9 cells. The effect of SPION treatment on cell cycle and DNA degradation was measured by flow cytometry after PI/Triton X-100 (PIT) staining (Appendix A). The measurements showed that SPION^LA-HSA^ did not significantly affect the amount of degraded DNA even with increasing concentration and in presence of external magnets. In contrast, treatment of all cell lines with SPION^LA^ above a concentration of 100 µg_Fe_/mL resulted in a significant increase in degraded DNA at the expense of double diploid DNA. Thus, the impact on DNA-degradation is clearly SPION-specific. In addition to viability experiments using flow cytometry, we investigated the effect of SPION treatment on proliferation using impedance- and optical-based life cell imaging (Figure 7, Appendix A). The effect of SPION^LA^ was similar for all cell lines. At a concentration of 50 µg_Fe_/mL, SPION^LA^ barely affected cellular growth, whereas cell growth was distinctly inhibited at 100 µg_Fe_/mL SPION^LA^. Further increase of SPION^LA^ concentration resulted in a rapid decrease of the growth curves, most likely representing cell death. In comparison, SPION^LA-HSA^ was substantially more biocompatible with all cell lines tested. In particular, for SCC-9 cells, cell proliferation was essentially affected only at the highest concentration (150 µg_Fe_/mL). At 50 and 100 µg_Fe_/mL, SPION^LA-HSA^ had no negative effect on cell proliferation. At the lowest concentration, cell growth was even enhanced, indicating a known effect of SPION-induced enhanced cell growth at lower concentrations that is reduced or abolished at increasing SPION concentrations due to cytotoxic effects [34].

### 3.7. Influence of SPIONs on Plasma Coagulation

Finally, after determining SPION-based cytotoxicity, we investigated the biocompatibility of the particles with human blood, which is crucial for potential clinical applications. An important parameter for testing the in vivo suitability of nanomaterials is their influence on the coagulation process. Therefore, SPION^LA^ and SPION^LA-HSA^ were added in ascending concentrations to fresh human citrate-stabilized plasma. After addition of activated partial thromboplastin time (aPPT) solution, coagulation was started with CaCl_2_ solution and clotting time was automatically monitored (Figure 8). The results demonstrated that SPION^LA^ increased the clotting time from 36 s to more than 50 s. In contrast, SPION^LA-HSA^ did not affect the clotting time regardless of the SPION concentration used, indicating a good biocompatibility for in vivo applications.

## 4. Discussion

We have previously demonstrated the biocompatibility of various SPIONs to different breast cancer cell lines [35]. In that study, SPION^LA-HSA^ showed excellent biocompatibility and SPION^LA^ only slightly enhanced cytotoxicity towards the selected tumor cells. In addition, the magnetic properties exhibited sufficient response to magnetic fields, which is a prerequisite for MDT. While the feasibility and efficacy of MDT have already been demonstrated in vitro, ex vivo, and in animal models [20,36,37,38,39], there are only very sparse studies on the use of SPIONs for MDT of head and neck tumors [40]. For the present study we therefore investigated the effects of selected SPIONs on different head and neck cancer cell lines. The work revealed several cell line-, particle-, concentration-, and magnetic force-dependent effects on cellular particle uptake and cytotoxicity. Since the investigated particle systems differed only by an additional protein corona of human serum albumin on SPION^LA-HSA^, which is not present on SPION^LA^, detected differences in cellular uptake and biocompatibility were mainly determined by the surface coating [22]. Compared to SPION^LA^, the stabilizing protein coating of SPION^LA-HSA^ led to a significantly lower cellular uptake and to a higher biocompatibility. Investigation of the physicochemical properties, in particular of SPION^LA^ particles, revealed that medium-based changes in surface coating can have a significant influence on the behavior of the particles (Table 1). Size determinations in different media, especially of SPION^LA^, showed strong agglomeration in FCS-free medium, while in FCS-containing medium the particles were stabilized, most likely by a forming protein corona [41]. Even if particles are stabilized in medium or in physiological fluids by a corona formation, it cannot be excluded that particles sediment over time, and the sedimentation rate may well differ between different nanoparticles [42,43,44]. Moreover, previous studies revealed a stronger sedimentation tendency of SPION^LA^ compared to SPION^LA-HSA^, even at different FCS concentrations [35]. Consequently, a stronger sedimentation most likely results in significantly increased particle-cell interactions that may lead to increased particle uptake and toxicity. This hypothesis offers a potential explanation of the dose-dependent effects on uptake and the various degrees of toxicity induced by the two SPION types in the cell lines studied.

The application of external magnetic fields additionally increases the sedimentation of SPIONs and thus the cellular interaction and uptake (Figure 5). At a SPION concentration of 100 µg_Fe_/mL SPION^LA-HSA^, the magnetic flux density of up to 380 mT applied here leads to a cell-dependent (7.9-fold to 13.8-fold) increase in cellular particles. Interestingly, after incubation with SPION^LA^ at the same concentration, the increase for iron detected by MP-AES was very similar (5.9-fold to 12.9-fold), indicating the presence of a similar magnetic susceptibility for SPION^LA^ and SPION^LA-HSA^. Thus, the presence of a magnetic field allows a 5.9-fold to 13.8-fold increase in particle uptake, which is similar for all cell lines and particles studied. However, there are significant cell type-dependent differences in absolute SPION uptake. While cells derived from tongue carcinoma (SCC-9) incorporated the most particles, closely followed by salivary gland carcinoma cells (A-253), the efficiency of cells derived from metastases of the hypopharynx (FaDu) and pharynx (Detroit 562) to take up particles was significantly lower. These cell-based differences were enhanced after application of a magnetic field. Thus, the observed variations in SPION uptake on a cellular level are not only due to the physicochemical properties of the coating but also to the different characteristics of head and neck cancer cells. Interestingly, despite an enhanced uptake, application of an external magnetic field revealed only a slight effect on cell viability in FaDu cells, whereas there was no obvious impairment of cell viability in Detroit 562, A-253, or even SCC-9 cells, which exhibited the highest SPION loadings. This is a surprising result, as increased SPION content can be expected to increase cytotoxicity. However, the magnetic field could induce the formation of larger aggregates that are only deposited on the cell membrane and do not enter the cell by active uptake mechanisms. Previously, a study addressing the magnetic settling of SPIONs demonstrated that magnetic fields lead to particle aggregation and a faster settling of magnetic nanoparticles [45]. Another study using small-angle neutron scattering confirmed the magnetic-field-induced transition of colloidal dispersed iron oxide nanoparticles to 1D chains and finally to 3D superstructures [46]. In addition, Min et al., investigated the effects of pulsed and constant magnetic fields on nanoparticle transport through cellular barriers [47]. While pulsed fields enhanced cellular uptake by increasing particle accumulation, the application of constant magnetic fields resulted in inhibition of cellular transport by the formation of aggregates larger than two micrometers, which exceeded the size of endocytic vesicles. This explains the differences in the quantification of cellular particle loading using different methods. During sample preparation for holotomographic imaging, cells had to be rigidly washed to avoid interference from any loosely bound or detached particle aggregates. Excessive particle deposition on the cell membrane would result in strong light refraction that would prevent detailed imaging of localization (exemplarily presented in Appendix A in the 2D and 3D images of SPION^LA^ -incubated cells). Flow cytometric quantification via SSc also underrepresents the particle amount because detached particle aggregates were gated out and apoptotic and necrotic cells, which may contain high particles quantities, were excluded from SSc determination to avoid including the increased SSc of dying cells (Figure 4). Similar to the two previously mentioned methods, quantification by MP-AES revealed higher uptake of SPION^LA^ compared to SPION^LA-HSA^ and increased SPION uptake of both particles in the presence of a magnetic field (Figure 5). This was due to the fact that the iron concentration of all cells, i.e., also apoptotic and necrotic cells, was included in the calculation. In addition, the samples were not washed as extensively, so that loosely bound cellular particle aggregates could also be taken into account. It should be emphasized that the fluorescence microscopic images shown do not always correspond to the quantification methods applied, which give very similar results among themselves. This is due to the fact that the microscopic images taken with a conventional fluorescence microscope were mostly focused on the cell nuclei and not on one of the planes in which particle-filled vesicles were located. Since fluorescence microscopy is difficult to completely visualize non-fluorescently labeled structures in the cell, other methods should be used for quantification where the 3-dimensional distribution of particles is not important, e.g., holotomographic microscopy, flow cytometric SSc analysis or atomic emission spectroscopy. Interestingly, in flow cytometric viability experiments and growth curve generation using live cell imaging and impedance-based technology, both SPION^LA-HSA^ and SPION^LA^ particles caused very little cytotoxicity at lower concentrations, even compared to control samples to which the appropriate volume of water was added. In a previous study, Gohlami et al., investigated the effect of different SPION systems on liver carcinoma cells and reported similar observations that SPIONs had a dose- and coating-dependent effect on cell viability [34]. At low SPION concentrations, the positive effect on cell viability was attributed to additional available iron as a metabolite. Because the expression of many molecules involved in cell cycle progression and proliferation is regulated by intracellular iron levels, tumor cells have increased iron requirements due to their increased proliferation rate. In contrast to the beneficial effects of moderate iron concentrations, iron deprivation has been reported to lead to cell cycle arrest in tumor cells [48].

Our study is consistent with Poller et al., who demonstrated that breast cancer cells incubated with SPION^LA-HSA^ had moderate SPION uptake with little effect on cytotoxicity, whereas SPION^LA^ showed a stronger effect on cellular uptake with a concomitant increase in cellular toxicity [35]. Notably, a comparison of the SPION uptake of breast cancer cell lines and head and neck tumor cell lines after 48 h of incubation with 50 µg_Fe_/mL each of SPION^LA-HSA^ and SPION^LA^ revealed very similar values (Appendix A). The detected iron content ranged from 0.4 to 1.0 pg/cell after SPION^LA-HSA^ incubation in the breast cancer cell lines and head and neck tumor cell lines and from 0.7 to 1.1.7 pg/cell after incubation with SPION^LA^. Only incubation of SPION^LA^ with primary human umbilical human endothelial cells (HUVECs), showed a comparatively higher particle uptake of 3.0 pg/cell. However, comparing the results is limited as breast cancer cells were not exposed to a magnetic field and incubated with the highest SPION concentration of 75 μgFe/mL. The results presented strongly suggest that despite the heterogeneity of cellular origin, the feasibility of MDT may be expanded to various entities beyond breast and head and neck cancer cell lines. However, SPION^LA^ exhibited substantial toxicity and decreased biocompatibility compared with SPION^LA-HSA^ (Figure 8). Based on this, SPION^LA-HSA^ is to be considered as the more appropriate particle to be used for MDT in head and neck cancer.

## 5. Conclusions

The present work provides new results on the interaction of SPIONs with head and neck cancer cell lines. The examined cell lines demonstrated a distinct ability to take up superparamagnetic iron oxide nanoparticles. Applying an external magnetic field does not impair cell growth and viability. Accordingly, the results indicate that head and neck cancers may serve as potential candidates for magnetic drug targeting using SPIONs.

## Figures and Tables

**Figure 1 nanomaterials-11-00726-f001:**
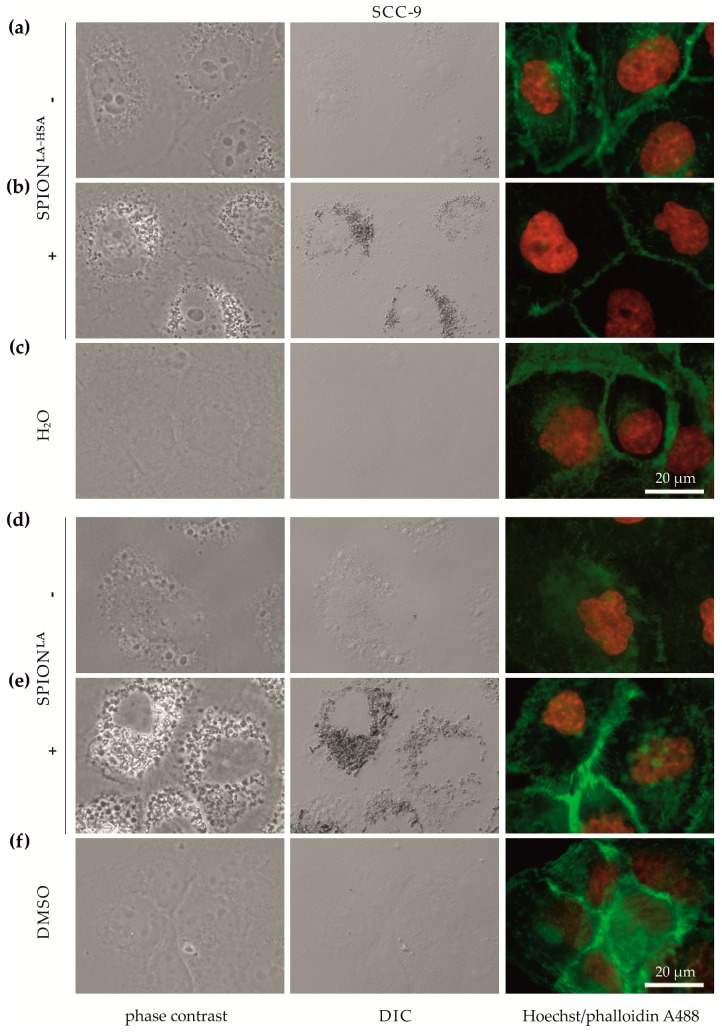
Optical imaging of SCC-9 cells incubated with SPIONs in the presence and absence of external magnets. Cells were treated with (**a**,**b**) SPION^LA-HSA^ and (**d**,**e**) SPION^LA^ (100 μgFe/mL) in the presence (+) or absence (−) of a magnet, or with the corresponding amount of (**c**) H_2_O or (**f**) DMSO (final concentration of 2%) for 48 h and visualized by phase contrast (first column), DIC (middle column) or fluorescence staining of nuclei (Hoechst 33342, red) and actin cytoskeleton (Alexa Fluor 488 Phalloidin, green) (last column). Abbreviations: SPION, superparamagnetic iron oxide nanoparticles; SPION^LA^, lauric acid-coated SPIONs; SPION^LA-HSA^, lauric acid- and human serum albumin-coated SPIONs; DIC, differential interference contrast; DMSO, dimethyl sulfoxide.

**Figure 2 nanomaterials-11-00726-f002:**
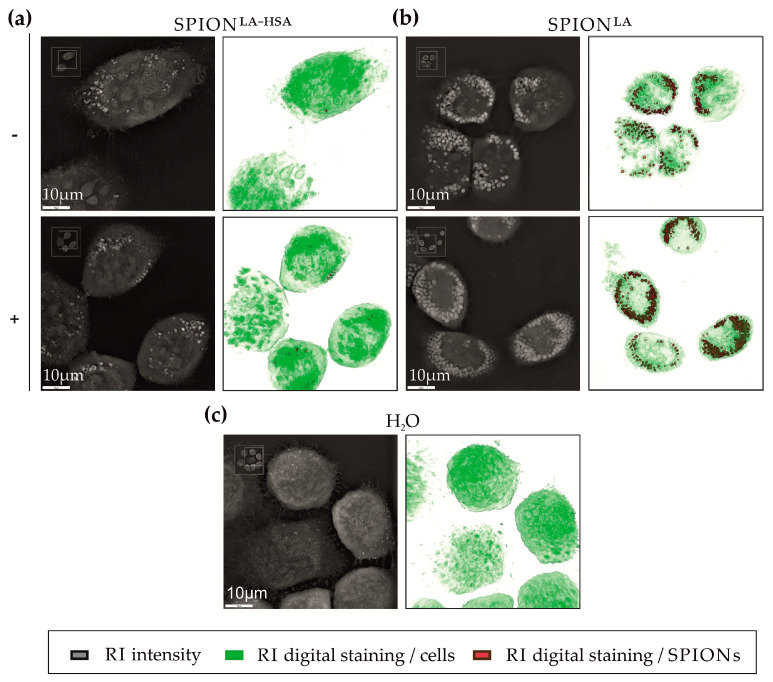
Holotomographic imaging of SCC-9 cells after incubation with SPIONs. Cells were incubated with SPIONs in the presence and absence of an external magnet. Cells were treated with unlabeled (**a**) SPION^LA-HSA^ and (**b**) SPION^LA^ (100 μgFe/mL) in the presence (+) or absence (−) of a magnet or with the corresponding amount of (**c**) H_2_O for 48 h and visualized by differences in refractive index. The 2D black and white images depict X-Y sections of the 3D RI composition. The 3D RI hologram is digitally colored, showing structures with low RI (cell nuclei, nucleoli and cytoplasm) in green and high RI (SPION clusters) in reddish-brown. The colored images show the top view of the 3D holograms. Abbreviations: SPION, superparamagnetic iron oxide nanoparticles; SPION^LA^, lauric acid-coated SPIONs; SPION^LA-HSA^, lauric acid- and human serum albumin-coated SPIONs; RI, refractive index.

**Figure 3 nanomaterials-11-00726-f003:**
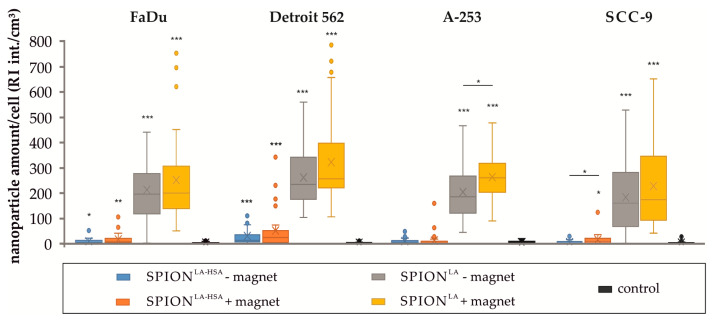
Nanoparticle amount quantified by RI analysis. The amount of SPIONs in different head and neck tumor cell lines treated with 100 μg_Fe_/mL SPION^LA^ or SPION^LA-HSA^ in the presence or absence of external magnets was determined by the presence and quantification of structures with high RI. Statistical significances are indicated with *, ** and ***. The respective confidential intervals are *p* ≤ 0.05, *p* ≤ 0.01, and *p* ≤ 0.001 and were calculated via *t*-test analysis. Asterisks on individual bars indicate the significance between SPION-treated cells and control samples and asterisks on connecting lines indicate the significance between samples incubated with SPIONs in presence or absence of magnets. Abbreviations: SPION, superparamagnetic iron oxide nanoparticles; SPION^LA^, lauric acid-coated SPIONs; SPION^LA-HSA^, lauric acid- and human serum albumin-coated SPIONs; RI int., refractive index intensity.

**Figure 4 nanomaterials-11-00726-f004:**
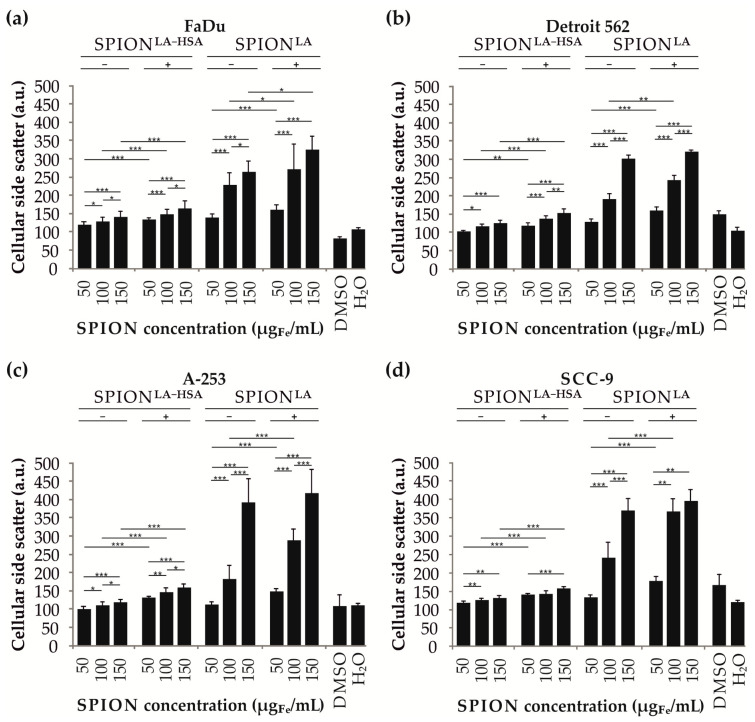
Evaluation of cellular nanoparticle uptake by changes in flow cytometric side scatter intensity. Head and neck tumor cell lines (**a**) FaDu, (**b**) Detroit 562, (**c**) A-253 and (**d**) SCC-9 were treated with 0, 50, 100 and 150 µg_Fe_/mL SPION^LA^ and SPION^LA-HSA^ for 48h. Control samples contained the appropriate amount of ultrapure water and toxicity controls DMSO (final concentration 2%). Bars show the arithmetic mean of SSc intensity. Error bars indicate the standard deviation of *n* = 4 with quadruplicates. Statistical significances are indicated with *, ** and ***. The respective confidential intervals are *p* ≤ 0.05, *p* ≤ 0.01, and *p* ≤ 0.001 and were calculated via Student’s *t*-test analysis. Abbreviations: SPION, superparamagnetic iron oxide nanoparticles; SPION^LA^, lauric acid-coated SPIONs; SPION^LA-HSA^, lauric acid- and human serum albumin-coated SPIONs; DMSO, dimethyl sulfoxide; SSc, side scatter; a.u., arbitrary unit.

**Figure 5 nanomaterials-11-00726-f005:**
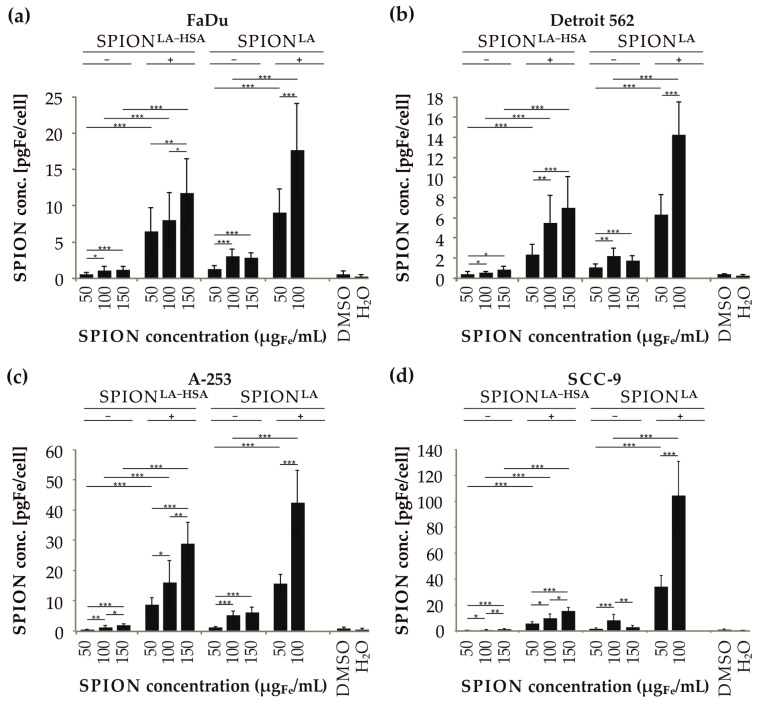
Quantification of the cellular nanoparticle load via MP-AES. The cell lines FaDu (**a**), Detroit 562 (**b**), A-253 (**c**) and SCC-9 (**d**) were spiked with 50, 100 and 150 µg_Fe_/mL SPION^LA-HSA^ or SPION^LA^ and further cultured either without (−) or on magnetic plates (+). Due to increased toxicity and low cell numbers, data sets obtained with 150 µg/mL SPIONLA in presence of magnets were removed for reliability reasons. Control samples contained the corresponding amount of H_2_O or DMSO at a final concentration of 2%. Data are expressed as mean standard deviation (n = 3 with technical triplicates). Statistical significances are indicated with *, ** and ***. The respective confidential intervals are *p* ≤ 0.05, *p* ≤ 0.001, and *p* ≤ 0.0001 and were calculated via Student’s *t*-test. Abbreviations: MP-AES, microwave plasma-atomic emission spectroscopy; SPION, superparamagnetic iron oxide nanoparticles; SPION^LA^, lauric acid-coated SPIONs; SPION^LA-HSA^, lauric acid- and human serum albumin-coated SPIONs; DMSO, dimethyl sulfoxide.

**Figure 6 nanomaterials-11-00726-f006:**
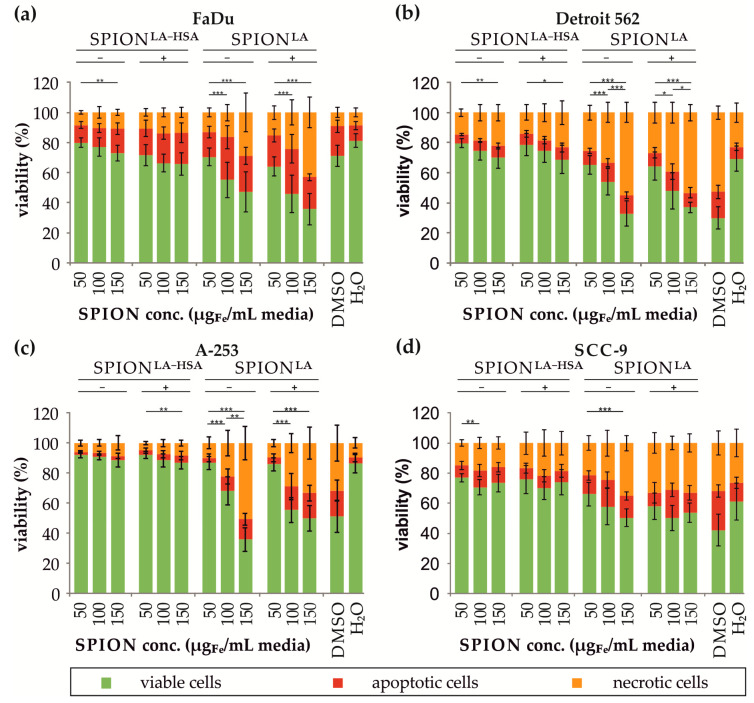
Viability of cell lines after SPION treatment in the presence or absence of magnets. Shown are the amount of viable (Ax- PI-), apoptotic (Ax+ PI-), and necrotic (PI+) cells after treatment of FaDu (**a**), Detroit 562 (**b**), A-253 (**c**) and SCC-9 cells (**d**) with SPION^LA^ or SPIONL^A-HSA^ at concentrations of 50, 100, and 150 µg_Fe_/mL without (-) or on (+) magnetic plates. Positive controls contain 2% DMSO and negative controls contain H_2_O instead of the SPION suspension. Data are expressed as mean standard deviation (n = 3 with technical triplicates). Statistical significance of viable cells at increased SPION concentration are marked with *, ** and ***. The respective confidential intervals are *p* ≤ 0.05, *p* ≤ 0.001, and *p* ≤ 0.0001 and were calculated using Student’s *t*-test analysis. Abbreviations: SPION, superparamagnetic iron oxide nanoparticles; SPION^LA^, lauric acid-coated SPIONs; SPION^LA-HSA^, lauric acid- and human serum albumin-coated SPIONs; Ax-PI-, annexin V-FITC-negative and propidium iodide-negative cells; Ax+PI-, annexin V-FITC-positive and propidium iodide-negative cells; PI+, propidium iodide-positive cells.

**Figure 7 nanomaterials-11-00726-f007:**
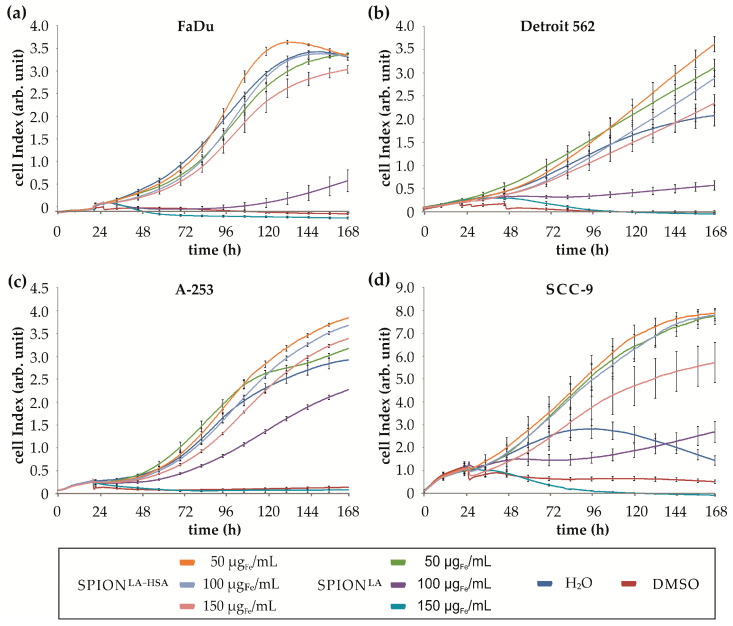
Impedance-based real-time analysis of cell growth. Growth curves of FaDu (**a**), Detroit 562 (**b**), A253 (**c**) and SCC-9 (**d**) cells were generated by monitoring the change in impedance after addition of SPION^LA^ or SPION^LA-HSA^ at concentrations of 50, 100 and 150 µg_Fe_/mL cell culture medium. The positive control contains 2% DMSO, and the negative control H_2_O instead of the SPION suspension. Data are expressed as standard error of the mean (*n* = 4 with 8-fold replicates). Abbreviations: SPION, superparamagnetic iron oxide nanoparticles; SPION^LA^, lauric acid-coated SPIONs; SPION^LA-HSA^, lauric acid- and human serum albumin-coated SPIONs; arb. unit, arbitrary unit.

**Figure 8 nanomaterials-11-00726-f008:**
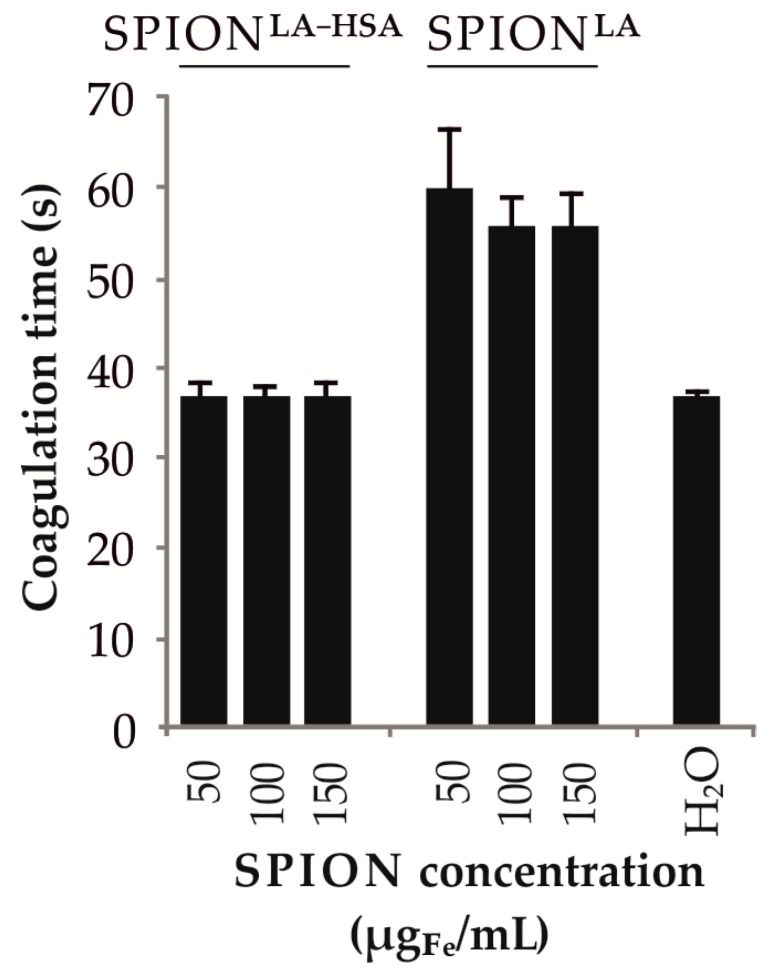
Influence of SPIONs on plasma coagulation. SPION^LA-HSA^ and SPION^LA^ were added to citrate-stabilized human blood at increasing concentrations. H_2_O served as control. After addition of aPTT solution, coagulation was induced by CaCl_2_ solution and coagulation time was automatically measured. Abbreviations: SPION, superparamagnetic iron oxide nanoparticles; SPION^LA^, lauric acid-coated SPIONs; SPION^LA-HSA^, lauric acid- and human serum albumin-coated SPIONs; aPPT, activated partial thromboplastin time.

**Table 1 nanomaterials-11-00726-t001:** Physicochemical characterization of superparamagnetic iron oxide nanoparticles. The hydrodynamic diameter of SPION^LA^ and SPION^LA-HSA^ were determined in H_2_O, media with 0% FCS and media with 10% FCS. Zeta potential and magnetic susceptibility were measured in water. Abbreviations: SPION, superparamagnetic iron oxide nanoparticles; SPION^LA^, lauric acid-coated SPIONs; SPION^LA-HSA^, lauric acid- and human serum albumin-coated SPIONs; FCS, fetal calf serum; Z ave. Ø, mean diameter.

Parameters	SPION^LA^	SPION^LA-HSA^
Z ave. Ø in H_2_O (nm)	46 ± 0.2	69 ± 0.4
Z ave. Ø in media with 0% FCS (nm)	2497 ± 278	68 ± 0.6
Z ave. Ø in media with 10% FCS (nm)	90 ± 0.2	50 ± 0.2
Zeta potential in H_2_O at pH ≈ 6.7 (mV)	−35 ± 0.4	−20 ± 0.8
Magnetic susceptibility normalized to 1 mg_Fe_/mL (in SI units)	7.27 × 10^−3^	7.37 × 10^−3^

## Data Availability

The data is available on reasonable request from the corresponding author.

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
