# Peer review of "Cellular SPION Uptake and Toxicity in Various Head and Neck Cancer Cell Lines"

_nanomaterials, 2021, doi:10.3390/nano11030726_

Round 1
Reviewer 1 Report
I write you in regards to manuscript entitled “Cellular SPION uptake and toxicity in various head and neck cancer cell lines” which you submitted to Nanomaterials.
As author notes in this report, this study might be useful information for Superparamagnetic iron oxide nanoparticles as vehicles for magnetic Drug Targeting. The paper provides interesting data but it still needs a considerable revision to be acceptable for Nanomaterials.
Minor comments
・Fig. 1, Supplemental Fig. S4, and S5: When SPIONLA-HAS and SPIONLA were incubated with a magnet, the fluorescence intensity of actin cytokeratin in the cytoplasm became weaker. What do you think about this?
・Page 17 Line 620-623: You mentioned the iron content in the breast cancer cell lines and the neck tumor cell lines. I think that the results for each cell lines should be shown.
Major comments
You have analyzed the amount of SPIONs using a variety of methods. However, I think that there is discrepancy in the results.
・Supplemental Fig. S3: The amount of SPIONLA was lower than that of SPIONLA-HAS. I think this result is different from other results.
・Fig. 2: In holotomographic imaging of SCC-9 cells after incubation with SPIONLA-HAS, the accumulation of SPIONLA-HAS is very little. On the other hand, SPIONLA-HAS is observed in optical imaging of SCC-9 cells (Fig.1).
Reviewer 2 Report
The article is concerned to the important for modern biomedicine problem of directed drug delivery using superparamagnetic metal or metal oxide nanoparticles. Superparamagnetic iron oxide nanoparticles (SPIONs) functionalized with chemotherapeutic agents were successfully directed into tumor regions via external magnetic fields, confirming the in vivo applicability of MDT. The present study determines SPION uptake, toxicity and biocompatibility in human head and neck tumor cell lines of the tongue, pharynx and parotid gland. The new and interesting results were obtained. The magnetic susceptibility measurements, microscopy, atomic emission spectroscopy, flow cytometry, and plasma coagulation the magnetic properties, cellular up take and biocompatibility of two different SPION types - coated with lauric acid or human serum albumin in the presence and absence of external magnetic fields were studied. It was shown Incubation of cells with lauric acid and human serum albumin-coated nanoparticles resulted in the substantial particle up take with low cytotoxicity. The uptake of lauric acid-coated nanoparticles was substantially increased but accompanied by higher toxicity. The presence of an external magnetic field significantly increased cellular uptake of both types of nanoparticles, although the cytotoxicity was not significantly increased. Finally, the results obtained, show the possibility of the potential use of SPIONs, developed in the work, as effective vector carriers for magnetic drug delivery in head and neck cancer therapy. The paper can be published in the present form after some minor revisions and improvements.

Reviewer 3 Report
This manuscript (Balk et al) describes two known types of SPIONs, each coated with lauric acid (LA) or LA-human serum albumin, studied for their cellular uptake and cytotoxicity in four head and neck cancer cell lines in vitro. This study appears an extension of similar studies (ref 21, 22), each reported by the lead authors for anticancer drug delivery. It is written clearly and reads well. However, its objectives are rather incremental in the scope and not strong enough in significance. Its data presented for their potential application in magnetic drug targeting (MDT) is inconsistent due to dependence on study methods and cell types. Overall, this reviewer feels that this manuscript is not strong enough in novelty and advances for its consideration and possible publication in Nanomaterials, one of the journals of growing impact in the field. Additional comments are provided for consideration for its improvement.
Comments:
- Introduction (page 1). Please describe clearly how this current study is of significant advance compared to prior works of relevance such as ref 21 and 22.
- Page 9 (lines 361-362). “enrichment of both SPION-LA and SPION-La/HSA in the presence of a magnetic field” While this statement is technically correct, these same NPs are similarly enriched in the absence of the magnetic field as indicated in Figure 2 and RI analysis in Figure 3. Only one cell line among four tested cells shows a small increase of statistical significance under the magnetic field.
- Page 10 (lines 396-399) and Figure 4. “higher when SPION-treated cells were incubated on magnet, indicating magnetically forced cellular accumulation. Thus, .. in very good agreement with the quantification.. (Figure 3)” This statement is also inconsistent with actual data presented in Figure 4 in which it shows no evidence for the difference of statistical significance between the presence and absence of the magnetic field in each tested cell line.
- Despite such lack of consistent data, magnetic field impact on the cellular uptake of SPIONs is overly stated in abstract, results and conclusion.
- Figure 3. Color coding for blue bars is not consistent with its legend.
- Page 9 (line 358) and Figure 2. Color coding in red is not clearly visible.
Round 2
Reviewer 3 Report
Reviewer's main concerns are addressed by presenting revised figures.